# Chemical Constituents, Antioxidant, and α-Glucosidase Inhibitory Activities of Different Fermented *Gynostemma Pentaphyllum* Leaves and Untargeted Metabolomic Measurement of the Metabolite Variation

**DOI:** 10.3390/antiox12081505

**Published:** 2023-07-27

**Authors:** Xuechun Zhang, Shi Li, Zhibin Zhang, Kin Weng Kong, Zhenxing Wang, Xiahong He

**Affiliations:** 1Key Laboratory for Forest Resources Conservation and Utilization in the Southwest Mountains of China, Ministry of Education, College of Life Science, Southwest Forestry University, Kunming 650224, China; zhangxuechun@swfu.edu.cn (X.Z.); lsfood980925@swfu.edu.cn (S.L.); zhenxing_wang@swfu.edu.cn (Z.W.); 2Key Laboratory of Protection and Utilization of Subtropical Plant Resources of Jiangxi Province, College of Life Science, Jiangxi Normal University, Nanchang 330022, China; 3Department of Molecular Medicine, Faculty of Medicine, Universiti Malaya, Kuala Lumpur 50603, Malaysia; kongkm@um.edu.my

**Keywords:** *Gynostemma pentaphyllum* leaves, probiotic fermentation, antioxidant capacity, α-glucosidase inhibitory activity, metabolomics

## Abstract

To assess the effects of microbial fermentation on *Gynostemma pentaphyllum* leaves (GPL), four probiotics were used to ferment GPL (FGPL) for 7 days. At different stages of fermentation, changes in the active components and biological activities of FGPL were determined. The findings suggest that short-term fermentation with probiotics can enhance both the content and bioactivity of active components in GPL. However, prolonged fermentation may lead to a decline in these aspects. Among them, the best effect was observed with SWFU D16 fermentation for 2 days. This significantly improved the total phenolic and total flavonoid content, antioxidant capacity, and inhibitory ability against α-glucosidase activity with an increase of 28%, 114.82%, 7.42%, and 31.8%, respectively. The high-performance liquid chromatography (HPLC) analysis results also supported this trend. Untargeted metabolomics analysis revealed metabolite changes between GPL and FGPL and the key metabolites associated with these functional activities. These key metabolites are mainly organic acids, flavonoids, carbohydrates, terpenoids, and other substances. KEGG analysis demonstrated that microbial metabolism in diverse environments and carbon metabolism were the most significantly enriched pathways. Among them, 3-(3-hydroxyphenyl) propanoic acid, d-glucose, gallic acid, gluconic acid, l-lactic acid, and l-malic acid were mostly involved in the microbial metabolism of diverse environmental pathways. In contrast, D-glucose, gluconic acid, and l-malic acid were mainly related to the carbon metabolism pathway. This study revealed the positive effect of probiotic fermentation on GPL and its potential metabolism mechanism, which could provide supporting data for further research.

## 1. Introduction

*Gynostemma pentaphyllum* (GP) is a plant belonging to the Cucurbitaceae family that is widely distributed in Northeast Asia and Southeast Asia. GP mainly grows on the hillsides of thickets, mountain slopes, and valleys at an altitude of 300–3200 m, where its leaf (GPL) is the main usable part [1]. As a popular folk medicine in Asia, GPL has a sweet and slightly bitter taste and is classified as “cold” in nature according to traditional Chinese medicine. In China, GPL is a common vegetable in daily life and can be used in various culinary forms (cooked, juiced, or stir-fried). Additionally, it is often consumed as tea, in drinks, and in other products. As early as 2002, the Chinese Health Commission issued the Notice of the “Ministry of Health on Further Regulating the Management of Raw Materials of Health Food” (No. 51 [2002]), which specified that GPL could be used as health food. In 2014, Shaanxi Province prescribed GPL as the local specialty food catalog (DB61/T 93.7-2014). Now GPL has been approved as a new type of resource food in China. Studies have indicated that GPL is rich in saponins, flavonoids, polysaccharides, and other active ingredients [2,3]. Due to its geographical distribution and significant biological activities, including antioxidant, antitumor, anti-inflammatory, antihyperglycemic, immunomodulatory, and cardiovascular protection, GPL is called “the Southern ginseng” [4,5,6,7,8,9]. Despite the numerous studies that have been conducted on this species, further research on GPL is still warranted due to its potential uses as an ingredient for medicinal, food, and healthcare products.

Fermentation technology is a significant food production and preservation technique. Under the fermentative action of microbiota, such as the synergistic actions of various hydrolytic enzymes produced by microorganisms, macromolecule substances in feedstocks are hydrolyzed into small molecules that are easily absorbed, improving the beneficial properties of the product [10,11]. At the same time, a variety of secondary metabolites are produced in fermented products, which cause changes in their chemical composition and properties [12]. Therefore, the changes in metabolites in fermented products also reflect the degree and quality of the fermentation. In addition, as important and special microorganisms, probiotics have been proven to have many health benefits, such as bacteriostatic, anticancer, and immune-enhancing [13,14]. With an in-depth understanding of their health effects, a vast array of probiotics has been used to ferment edible or medicinal plants. Despite the numerous health advantages of probiotic fermentation, the changes in functional activities and metabolites during the fermentation process and their underlying mechanisms are understudied and unclear.

With the progress of different chromatographic techniques, contemporary studies have implemented numerous methods for extensively analyzing chemical components and metabolites in the food system. For instance, liquid chromatography-tandem mass spectrometry (LC-MS/MS) and gas chromatography-tandem mass spectrometry (GC-MS/MS) [15,16]. These holistic approaches can accurately and rapidly detect the small molecules of foods/drugs. Therefore, they contribute significantly to more detailed studies on the fermentation process.

Therefore, based on the excellent properties of GPL and the significant advantages of fermentation techniques, we used different probiotics to ferment GPL. We measured and compared their polyphenol composition, antioxidant activities, and α-glucosidase inhibitory activity in different fermentation stages. Additionally, we analyzed the metabolic profiles of GPL before and after fermentation and revealed its possible metabolic pathways. This study explores the enhanced effect and mechanism of action of probiotic fermentation technology against GPL and could provide references for the further utilization of GPL.

## 2. Materials and Methods

### 2.1. Materials and Reagents

*Fresh GPL* was collected from Xishuangbanna, Yunnan Province, China, in January 2020. *Lactobacillus plantarum* (ATCC 8014), *Lactobacillus casei* (ATCC 334), and *Lactobacillus rhamnosus* (ATCC 53013) were purchased from the Guangdong Province General Microbiological Culture Collection Center and the American Microbial Species Preservation Center, respectively. *Lactobacillus plantarum* (SWFU D16) was isolated by ourselves.

Folin–Ciocalteu reagent, gallic acid, rutin, *sodium nitrite* (NaNO_2_), aluminum nitrate (Al(NO_3_)_3_), methanol, potassium persulfate, ferrous sulfate (FeSO_4_), ferric chloride hexahydrate (FeCl_3_·6H_2_O), sodium carbonate (Na_2_CO_3_), trisodium phosphate (Na_3_PO_4_), 2,2′-azino-bis (3-ethylbenzothiazoline-6-sulfonic acid) (ABTS), 1,1-diphenyl-2-picrylhydrazyl (DPPH), 2, 4, 6-tri(2-pyridyl)-s-triazin (TPTZ), and other chemicals were purchased from Aladdin (Shanghai, China). Chromatographic acetonitrile was purchased from Merck (Darmstadt, Germany). P-nitrophenyl-α-D-glucopyranoside (PNPG) and α-glucosidase (G5003) were purchased from Sigma-Aldrich (St. Louis, MO, USA). Catechins, chlorogenic acid, and other standards were purchased from Yuanye Bio-Technology (Shanghai, China).

### 2.2. Fermentation and Sample Preparation

The fresh GPL was properly washed with distilled water and dried until a constant weight was reached at 50 °C. After drying, the samples were crushed and sieved through a 50-mesh sieve before being preserved in self-sealing bags and stored at −2 °C.

The method of preparing the fermented GPL (FGPL) was slightly modified from the previous method [17]. Briefly, the GPL powder (2 g) was accurately weighed in a conical flask, and then 50 mL of distilled water was mixed in it. Immediately, the final volume was adjusted by adding distilled water to 100 mL and sterilized at 12 °C for 15 min. After cooling to room temperature, 10 mL of a fresh bacterial solution was added at 1–5 × 10^7^ CFU/mL. Then, the fermentation was performed at 37 °C and lasted 7 days. The fermentation broths were harvested daily, then separated by centrifugation at 5000× *g* for 10 min at 25 °C. The supernatant was collected and stored at −80 °C for the subsequent determinations.

### 2.3. Fermentation pH

The pH values were determined at room temperature using a pH meter (Hanna Instrument^®^, Ann Arbor, MI, USA). Briefly, the fermentation broth is collected once every 24 h and then centrifuged at 5000× *g* for 10 min at 25 °C. Finally, the supernatant is collected for analysis.

### 2.4. Determination of Total Phenolic Content

The total phenolic content (TPC) was measured by the Folin–Ciocalteu method with slight modifications [18]. 125 μL of Folin–Ciocalteu reagent (10%, *w*/*v*) and 100 μL of Na_2_CO_3_ solution (7.5%, *w*/*v*) were added to 50 μL of supernatant and mixed. At room temperature, the mixture was incubated for 30 min. Then, the sample absorbance was measured at 765 nm. Gallic acid (0.1–1 µmoL/mL) was used as a standard. The final results were further calculated based on the ratio of the TPC on Day n to that of Day 0 and expressed as relative TPC (%).

### 2.5. Determination of Total Flavonoid Content

The total flavonoid content (TFC) of the supernatant was examined by the sodium nitrite–aluminum nitrate colorimetric method with some modifications [19]. 20 μL of NaNO_2_ (3%, *w*/*v*) was added to 40 μL of the sample and incubated for 6 min. A further 20 μL of Al(NO_3_)_3_ (6%, *w*/*v*) was then added and held for a further 6 min. Subsequently, 140 µL of NaOH at a concentration of 4% (*w*/*v*) and 60 µL of methanol at a concentration of 70% were added and incubated for 15 min at room temperature. Finally, the absorbance of the sample solution was measured at a wavelength of 510 nm. 0.05–0.5 µmoL/mL of rutin was used as a standard curve. The results were further calculated based on the ratio of the TFC on Day n to that of Day 0 and expressed as relative TFC (%).

### 2.6. Determination of DPPH· Scavenging Capacity

The DPPH radical scavenging capacity of the sample was assessed according to the reported method [20]. A total of 100 μL of 0.15 mM DPPH solution was added to 100 μL of supernatants and incubated in the dark for 30 min. Then, the absorbance of the mixture was measured at 517 nm. The results were expressed as relative DPPH· scavenging capacity (%), which was calculated according to the ratio of the DPPH scavenging capacity on Day n to that of Day 0, and the DPPH scavenging rate was as follows:(1)$$\mathrm{Scavenging}\;\mathrm{rate}\;(\%)=\;(1-\frac{A_{\mathrm{sample}}-A_{\mathrm{control}}}{A_{\mathrm{blank}}})\;\times \;100$$
where *A_sample_* = absorbance of the sample group; *A_control_* = absorbance of the control group; and *A_blank_* = absorbance of the blank group.

### 2.7. Determination of ABTS·+ Scavenging Capacity

The ABTS radical scavenging capacity was determined according to a slightly modified method reported previously [21]. Briefly, 200 µL of freshly prepared ABTS·+ working solution was added to 50 µL of the appropriately diluted sample and incubated for 6 min. The absorbance at 734 nm was read. The results were also expressed as the relative ABTS·^+^ scavenging capacity (%). Equation (1) of the clearance rate is the same as previously mentioned.

### 2.8. Determination of Ferric Reducing Antioxidant Power

Briefly, 250 µL of the freshly prepared FRAP solution was added to 50 µL of the sample in a 96-well plate. After incubation at 37 °C for 10 min, the absorbance value was determined at 593 nm wavelength, and the standard curve was constructed using FeSO_4_ [22]. The FRAP value of the sample was converted to the mg of FeSO_4_ equivalent per gram of sample (mg FeSO_4_/g sample), and the final results were equally expressed as the relative percentage (%).

### 2.9. Determination of α-Glucosidase Inhibitory Activity

According to the previously reported method [23], 25 μL of *α*-glucosidase solution with a concentration of 0.1 U/mL was mixed into 50 μL of the sample. After 37 °C incubation for 10 min, 50 μL of 5 mM PNPG was added and incubated at 37 °C for 15 min. Finally, 100 μL of Na_2_CO_3_ (0.2 M) was added to end the reaction. An absorbance detector measured the absorbance value at 405 nm. The results were expressed as the relative α-glucosidase inhibitory activity (%) according to the ratio of the α-glucosidase inhibitory rate on Day n to that of Day 0, and the calculation was conducted using the previous Equation (1).

### 2.10. HPLC Analysis

The HPLC analysis was analyzed using an Agilent 1260LC system (Agilent Technologies, Santa Clara, CA, USA) [24]. Before injection, all samples were filtered by 0.22 µm nylon syringe filters. The C18 reversed-phase analytical column (250 mm × 4.6 mm, 5 µm, Greenherbs Science and Technology, Beijing, China) was maintained at 25 °C. Mobile phase A consisted of water containing 0.1% formic acid, and mobile phase B consisted of acetonitrile. The liquid chromatography gradient was as follows: 2–8% B in 0–12 min, 8–13% B in 12–15 min, 13–18% B in 15–30 min, 18–30% B in 30–50 min, 30–50% B in 50–60 min, 50–70% B in 60–70 min, 70–90% B in 70–80 min, 90–100% B in 80–85 min, and 100–2% B in 85–90 min. Each filtered 10 μL sample was determined in the UV scan range of 200–400 nm. The compounds were identified and quantified by comparing the retention times and peak areas of the authentic flavonoid and phenolic acid standards (gallic acid, catechin, chlorogenic acid, epicatechin, dihydromyricetin, and epicatechin gallate). For detailed information, refer to Appendix A.

### 2.11. Untargeted Metabolomics Analysis

The samples were treated according to the reference method for further UPLC-MS/MS untargeted metabolomic analysis [25]. For liquid chromatographic separation, an ultra-performance liquid chromatography (UPLC) system (Agilent 1290 infinity, Agilent Technologies, Santa Clara, CA, USA) with a C-18 column (Waters, ACQUITY UPLC BEH C-18 1.7 μm, 2.1 mm × 100 mm column) was used. The injection volume, flow rate, and column temperature were 2 μL, 0.4 mL/min, and 40 °C, respectively. Mobile phase A consisted of a 25 mM ammonium acetate solution containing 0.5% formic acid, and mobile phase B consisted of methanol. The gradient elution was programmed as follows: 5% B in 0–0.5 min, 5–100% B in 0.5–10 min, 100% B in 10–12 min, 100–5% B in 12.0–12.1 min, and 5% B in 12.1–16 min.

Mass Spectrometry Conditions: An AB Sciex 6600 Triple TOF instrument (AB Sciex, Path Framingham, MA, USA) was used, and all samples were measured in positive and negative ion modes. The Ion Source Gas1 was 60, Ion Source Gas2 was 60, the curtain gas was 30, the source temperature was 600 °C, the IonSpray Voltage Floating (ISVF) was 5500 V, the TOF MS scan *m*/*z* range was 60–1000 Da, the product ion scan *m*/*z* range was 25–1000 Da, the declustering potential (DP) was ±60 V, the collision energy was 35 ± 15 eV, the TOF MS scan accumulation time was 0.20 s/spectra, and the product ion scan accumulation time was 0.05 s/spectra. The mass spectrometry data acquisition was carried out with the Information Dependent Acquisition (IDA) method in high-sensitivity mode, where isotopes within 4 Da were excluded and the number of candidate ions to monitor per cycle was 10. Moreover, quality control (QC) samples were used to verify the system’s reliability. Metabolites were identified by comparing their MS/MS spectra with an in-house database provided by Applied Protein Technology Co., Ltd., Shanghai, China.

### 2.12. Statistical Analysis

All the experiments were performed in triplicate, and the results were presented as means ± standard deviations (SD). Microsoft Excel, Origin 2018, and IBM SPSS Statistics V.25.0 were used for statistical analysis. The significance of the data was analyzed using Duncan’s test, and 0.05 was considered significant. The principal component analysis (PCA) and correlation analysis were conducted using the R language package (R 4.0.3).

## 3. Results and Discussion

### 3.1. Fermentation pH

The variation of pH values in fermentation resulted from the microorganism’s metabolism, which also greatly represents the degree of fermentation. Figure 1(a1–a4) shows the changes in pH values in the different FGPL at different fermentation stages. There was a dramatic decrease in the early stage (1–3 days), indicating that all four probiotics were strongest during the early fermentation stage and possessed the best fermentation performance. As the fermentation continued, the effect of probiotic fermentation gradually decreased, leading to a constant pH value in the later stages owing to the depletion of nutritional components [25]. Among them, ATCC 8014 and SWFU D16 showed a more drastic decrease in the early stage, indicating that their fermentation capability was higher than that of the others.

### 3.2. Total Phenolic Content and Total Flavonoid Content

The changes in TPC and TFC in FGPL during the probiotic fermentation are shown in Figure 1(b1–b4,c1–c4). Significant differences can be observed among the trends in different strains and their results at different fermentation time points. The TPC increased after being fermented with ATCC 8014, ATCC 334, and SWFU D16 during short-term fermentation (1–3 days) in relation to their unfermented samples, as shown in Figure 1(b1–b4). Surprisingly, SWFU D16 and ATCC 53013 showed more fluctuating changes. Specifically, the maximum value of SWFU D16 was reached on day 3 of fermentation, and the maximum value of ATCC 53013 was reached on day 6 of fermentation, accounting for 128% and 197.5% of their unfermented samples, respectively. On the other hand, the TFC indicated a decreasing trend overall in Figure 1(c1–c4). However, a tendency to increase was observed for ATCC 334 and SWFU D16 during short-term fermentation. They achieved the maximum values on day 2 (115.28% and 214.82%, respectively), which was similar to that of fermented mulberry leaf reported previously [26].

Fermentation can cause the release of phenolic and flavonoid compounds from complex plant matrixes via different enzymatic reactions, leading to an increase in TPC and TFC in the fermented product. Additionally, fermentation can generate various new bioactive compounds [25]. However, these compounds may undergo further metabolic or chemical conversion and degradation as the fermentation time increases. Fermentation has been found to significantly increase the TPC and TFC in common buckwheat and Tartary buckwheat. However, with prolonged fermentation, the rutin content in fermented common buckwheat slightly reduces, which leads to a further reduction in the TFC [27]. In general, for FGPL, the TPC and TFC could increase to different degrees after a short-term fermentation, but they decreased throughout the fermentation.

### 3.3. Antioxidant Capacity

Due to the different mechanisms of antioxidant action, three assays were used to assess the antioxidant capacity of the samples [28,29], and the results are shown in Figure 1(d1–d4,e1–e4,f1–f4). Taken as a whole, all three antioxidant capacities generally exhibited a downward trend after 7 days of fermentation, and the changing trend was different on different antioxidant indexes for the same probiotic. But we were pleasantly surprised to find that certain probiotics could improve these activities to various degrees, such as ATCC 8014 and SWFU D16 on the second day, which were similar to TPC and TFC. Previous studies have shown similar results, such as that *L. plantarum* could increase the phenolic content of blueberries and improve their antioxidant and antitumor activities in cervical cells [30]. Additionally, *L. rhamnosus GG* and *L. plantarum-1* were shown to enhance the in vitro antioxidant capacities of blueberry pomace. This might partially be due to the increase in its TPC and TFC [31].

To illustrate the relationship between these indicators clearly, the correlation heat map, correlation network diagram, and PCA diagram were conducted and are shown in Figure 2. The correlation heat map showed that TPC and TFC had a good correlation with antioxidant capacity, which was also confirmed by the correlation network diagram and PCA diagram. The above results indicated that phenolics and flavonoids were the primary contributors to antioxidant capacity. This also illustrated the reason why there were similar trends in the changes in TPC, TFC, and antioxidant capacities. 

### 3.4. α-Glucosidase Inhibitory Capacity

α-Glucosidase is an exocyclic enzyme that belongs to the oligosaccharide hydrolases. It can hydrolyze the 1,4-α-glycosidic linkages of oligosaccharides and plays an important role in the carbohydrate metabolism of humans, animals, plants, and microorganisms. Thus, it is one of the medication targets in diabetic management [32]. As shown in Figure 1(g1–g4), ATCC 8014, ATCC 334, and SWFU D16 could enhance the α-glucosidase inhibitory capacity of GPL to different degrees in a short period of time (1–4 days). Among them, ATCC 334 reached the maximum value on the first day, which was 155.6% of that of the unfermented sample. ATCC 8014 and SWFU D16 reached their maximums (129.5% and 131.8%) on the fourth and third days, respectively. Although there was a slight decrease for ATCC 53013, there were no significant changes throughout the fermentation process. A similar study also demonstrated that different strains showed various promotion effects on the antioxidant and α-glucosidase inhibitory capacities of blueberry juice compared to spontaneous fermentation [33]. The above results suggested that the fermentation strain was the key to improving the functional activity of the fermented product. Earlier studies reported a positive relationship between TPC and the capacity to inhibit α-glucosidase [34]. This was also observed in this study, as shown in the correlation heat map and network map (Figure 2), where TPC was significantly associated with the inhibitory capacity of α-glucosidase.

### 3.5. HPLC Analyses

The results of the above analysis suggested that some active compounds in GPL were likely to be degraded and transformed during the fermentation process. Therefore, using HPLC, six compounds were identified and quantified in fermentation with different strains at different fermentation times. These compounds are gallic acid, catechin, chlorogenic acid, epicatechin, dihydromyricetin, and epicatechin gallate, respectively (Figure 3).

According to the results of TPC and TFC, the samples from the second day showed higher increase rates, so they were selected for analysis. As shown in Figure 3A, their approximate chromatogram peak shapes at 280 nm were the same for different strains, but the peak heights and peak areas were different. Of those, peak No. 2 (catechin) (details are shown in Table 1) of SWFU D16 was significantly higher than that of other probiotics. In contrast, all strains had no marked increase in peak No. 3 (chlorogenic acid) compared to unfermented samples. Interestingly, ATCC 334 and ATCC 53013 chromatograms showed more small chromatographic peaks between 5–10 min. In addition, there was a strong peak at approximately 18.5 minutes in unfermented GPL. It was observed that this peak greatly decreased or disappeared after fermentation by each strain, although it was not identified by matching standard compounds in this study. The above results suggest that the compounds in GPL were degraded to varying degrees during probiotic fermentation while also producing some new compounds. Similar results have been reported previously, in which fermentation of *L. plantarum* could greatly change the contents of phenolic components in dried longan pulp, such as gallic acid, vanillic acid, and 4-methyl catechol [35].

Similarly, given the better-enhanced effects of *L. plantarum* SWFU D16 on TPC, TFC, FRAP, and α-glucosidase inhibition capacity, the chromatograms at 280 nm of FGPL fermented with SWFU D16 at different fermentation stages were analyzed (Figure 3B). With the prolonged fermentation time, peak 1 (gallic acid) gradually decreased and fell to a low level in the later stage of fermentation. Peak 2 (catechin) began to increase on the second day and then gradually decreased, while peak 3 (chlorogenic acid), peak 4 (epicatechin), peak 5 (dihydromyricetin), and peak 6 (epicatechin gallate) had no obvious change. Another strain with a well-promoting effect on TPC, TFC, and α-glucosidase inhibition capacity, *L. casei* ATCC 334, exhibited a similar pattern in its chromatograms (Figure 3C). Compounds with large variations in the fermentation process, such as catechin, have been reported to exert good antioxidant activity and antitumor and antibacterial capabilities [36]. Thus, this might cause changes in the functional activities of GPL after fermentation. The above results fully demonstrated that different probiotics had different effects on the chemical components in GPL through different modes of action, such as degradation and transformation, affecting their functional activities.

### 3.6. Metabolomic Analysis

According to the above comprehensive analysis results, FGPL (fermented with SWFU D16 for 2 days) showed higher active ingredient contents and functional activities. Therefore, it was used for untargeted metabolomics analysis to investigate the metabolite changes between GPL and FGPL in depth. From the representative base peak chromatograms (BPC) in Figure 4A–D, the vast majority of the peak signal intensity of FGPL was lower when compared with GPL. This indicated that various metabolite contents in GPL decreased after fermentation.

Using the UHPLC-triple-TOF-MS/MS method, the structure of metabolites in biological samples was identified by matching the molecular mass of metabolites in the database (molecular mass error within <10 ppm), secondary fragmentation spectra, retention time, and other information (please refer to the Supplementary Information for details). In this study, 909 metabolites were identified, of which 514 and 395 were detected under positive and negative ion modes, respectively (details are shown in Appendix A). To easily visualize the classification of these metabolites, the corresponding pie charts were drawn, as shown in Figure 4E,F. Colors represent various metabolite species; the pie chart areas represent metabolite proportions. As shown in the figure, most of the metabolites in the superclass belong to lipids and lipid-like molecules, organoheterocyclic compounds, phenylpropanoids and polyketides, benzenoids and organic acids, derivatives, etc. The class included prenol lipids, benzene and substituted derivatives, carboxylic acids and derivatives, organooxygen compounds, fatty acyls, flavonoids, etc. At the same time, the subclass contained amino acids, peptides, analogs, carbohydrates and carbohydrate conjugates, terpene glycosides, flavonoid glycosides, benzoic acids, derivatives, amines, and fatty acyl glycosides.

A discriminant analysis of the metabolite composition before and after fermentation was also conducted using PCA. As shown in Figure 4G,H, the metabolites of the two groups have significant differences in both ion modes, indicating fermentation’s great influence on the metabolites of GPL. Volcanoes (Figure 4I,J) were constructed according to *p*-values < 0.05 and fold change (FC) ≥ 1.5 to better display the differential metabolites. In the positive ion mode, 29 metabolites were significantly altered, with 4 metabolites upregulated and 25 metabolites downregulated, respectively. In the negative ion mode, there were 40 significant differential metabolites. Of these, the upregulated metabolites were 12, and the downregulated metabolites were 28. At the same time, to generate hierarchical cluster heatmaps, the metabolites with the variable importance in projection (VIP) value (obtained by OPLS-DA) greater than one and the *p*-value from the results of *t*-tests less than 0.05 were selected. As shown in Figure 4K,L, there were more downregulated than upregulated metabolites, which indicated that SWFU D16 caused greater degradation of the chemical components in GPL. In addition, these differential metabolites mainly included organic acids, flavonoids, carbohydrates, terpenoids, and other substances, and these metabolites might be the main substance factors for the functional activities of GPL. For example, catechol has been reported to have good antioxidant and anti-inflammatory activities [37], and important miscellaneous derivatives like 9h-pyrido [3,4-b] indole show good antitumor activity in the organism [38].

Spearman’s correlation analyses were conducted to investigate the association between metabolites and functional activities, and the results were shown in Figure 5A–D using heatmaps and network diagrams. For some metabolites significantly associated with biological activities, we further mapped their chemical structural formulas in correlation network diagrams. Among them, FRAP was mainly positively correlated with downregulated metabolites. On the other hand, DPPH and ABTS radical scavenging activities displayed significant associations with more upregulated metabolites. This explained the change in the activities of GPL after fermentation. Surprisingly, α-glucosidase inhibition capacity showed weak correlations with nearly all metabolites, which indicated that it was the integrative action of multiple metabolites.

A KEGG pathway analysis (www.kegg.jp/kegg/pathway.html (accessed on 20 February 2023)) was performed for the differential metabolites to further explore the mechanism of the probiotic fermentation process. Through the KEGG analysis, 13 metabolic pathways were enriched. From the pathway network diagrams in Figure 5E, microbial metabolism in diverse environments and carbon metabolism were the most enriched pathways based on the number of enriched metabolites. Among them, microbial metabolism in diverse environments refers to the ability of microorganisms to adapt and metabolize in a wide range of habitats and conditions. Microbes exist in various environments, such as soil, water, plants, animals, and the human gut, and their metabolism is influenced by the unique characteristics of each ecosystem. The carbon metabolism pathway is crucial to microbial metabolism in probiotic fermentation. In probiotic fermentation, microorganisms metabolize sugars and carbohydrates through the carbon metabolism pathway to produce energy and generate metabolites. Meanwhile, as the most important basic metabolism in plants, carbon metabolism can provide essential energy for plant life activities [39,40]. Furthermore, the heatmap plots were produced to clearly display the metabolite level differences in the key network nodes of the two pathways with the most enriched metabolic pathways. Among them, 3-(3-hydroxyphenyl) propanoic acid, d-glucose, gallic acid, gluconic acid, l-lactic acid, and l-malic acid were mostly involved in the microbial metabolism of diverse environmental pathways. In contrast, D-glucose, gluconic acid, and l-malic acid were mainly related to the carbon metabolism pathway. Overall, many metabolites were not working alone and were involved in multiple functional activities and metabolic pathways.

## 4. Discussion

GP is a commonly found plant in China with high health benefits and medicinal value for human consumption [3]. Fermentation technology has been used in human history for over a thousand years. Previous studies have shown that probiotic fermentation can enhance plants’ nutritional value and functional activity [10]. 

In this study, to assess the effects of microbial fermentation on GPL, four probiotics were used to ferment GPL (FGPL) for 7 days, and changes in the active components and biological activities of FGPL at different fermentation stages were determined. The results showed that after 7 days of fermentation, the four probiotics had significantly different effects on GPL’s TPC, TFC, antioxidant capacity, and α-Glucosidase inhibitory capacity. Overall, they showed a trend of an initial increase followed by a decrease. Later, through quantitative analysis with HPLC and six compounds, it was found that as the fermentation time was extended, the content of gallic acid gradually decreased, the content of catechin showed a trend of an initial increase followed by a decrease, and the content of the other four compounds remained basically unchanged. This also revealed that different probiotics have different effects on the chemical components of GPL, such as degradation and transformation, thereby affecting its functional activity. During the fermentation process, probiotics can produce various hydrolytic enzymes. This can break down the cell wall components of plants, such as cellulose, pectin, and lignin, causing the decomposition of macromolecules and enhancing nutrients and active substances [25]. The organic matter of GPL itself also provides carbon and nitrogen sources for the fermentation of probiotics, which collectively enable short-term FGPL to exhibit positive functional activities. However, the fermentation process continuously consumes nutrients, and the enzymes produced by probiotics can also inhibit cellulase activity and decompose the compounds fermented by probiotics, leading to decreased functional activity of GPL after long-term fermentation [25].

Previous studies have also shown that short-term fermentation (1–3 days) can significantly improve the active components in *Perilla* leaves and the inhibition ability of α-glucosidase, etc. However, long-term fermentation can reduce the active components and biological activities [25]. Similar results have also been reported by Ru et al. [17]. These research results indicate that appropriate fermentation time is an effective method to promote the bioactivity of plant samples. 

We analyzed the changes in metabolites before and after FGPL had been fermented with SWFU D16 for 2 days based on the UHPLC-triple-TOF-MS/MS method to further explain the mechanism of metabolic products. The data showed that the main differences between GPL before and after fermentation were in the primary and secondary metabolites, such as flavonoids and fatty acids. By performing KEGG pathway analysis on significant differential compounds, we discovered 13 signaling pathways, where we found out that microbial metabolism in diverse environments and carbon metabolism were the most likely metabolic pathways. Microbial metabolism in diverse environments mainly involves substance transformation through glycolysis and the TCA cycle, while carbon metabolism includes the TCA cycle, the pentose phosphate pathway, and glycolysis.

## 5. Conclusions

This study clearly demonstrated the changes in the active ingredients and biological activity of GPL at different fermentation stages. Interestingly, a suitable probiotic (SWFU D16) was also found for FGPL, and its mechanism of action was deciphered. The results could provide a reference for the development and utilization of GPL. They will provide new insights and ideas for the biotransformation of the key active components in food or traditional Chinese medicine. This research is still in its initial stage, requiring further intensive studies, such as strain screening, process optimization, and in-depth functional studies, to be conducted in this respect.

## Figures and Tables

**Figure 1 antioxidants-12-01505-f001:**
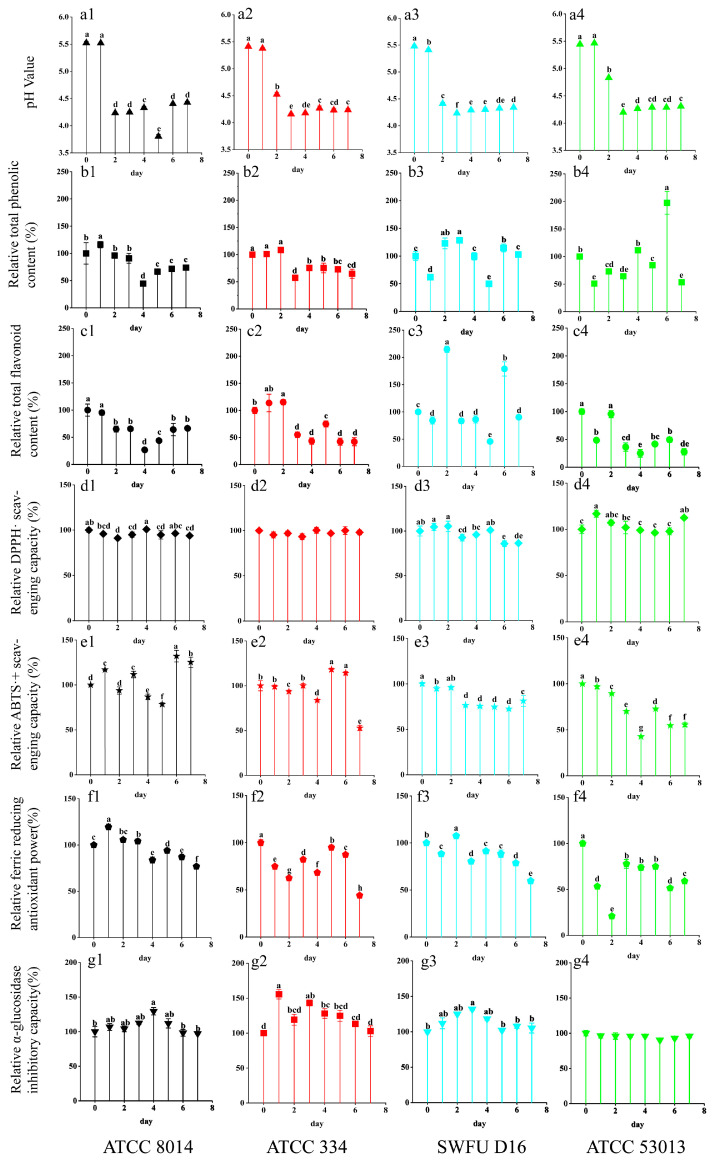
Changes in the pH value, total phenolic content (TPC), total flavonoid content (TFC), DPPH· scavenging capacities, ABTS·^+^ scavenging capacities, ferric reducing antioxidant power (FRAP), and α-glucosidase inhibitory capacity of GPL at different fermentation stages with different probiotics. (**a1**–**a4**) Changes in the pH value; (**b1**–**b4**) Changes in the TPC; (**c1**–**c4**) Changes in the TFC; (**d1**–**d4**) Changes in DPPH· scavenging capacity; (**e1**–**e4**) Changes in ABTS·^+^ scavenging capacity; (**f1**–**f4**) Changes in FRAP; (**g1**–**g4**) Changes in the α-glucosidase inhibitory capacity of GPL during probiotic fermentation; the labels (**a1**–**g4**) represent the identification numbers of each indicator chart, and the following numbers 1, 2, 3, and 4 correspond to the following strains: (1) ATCC 8014 (*L. plantarum*), (2) ATCC 334 (*L. casei*), (3) SWFU D16 (*L. plantarum*), and (4) ATCC 53013 (*L*. *rhamnosus*). The symbols “a, b, c, d, ab, cd, et al.” at the top of the bar graph represent the significance analysis of each index during the fermentation process.

**Figure 2 antioxidants-12-01505-f002:**
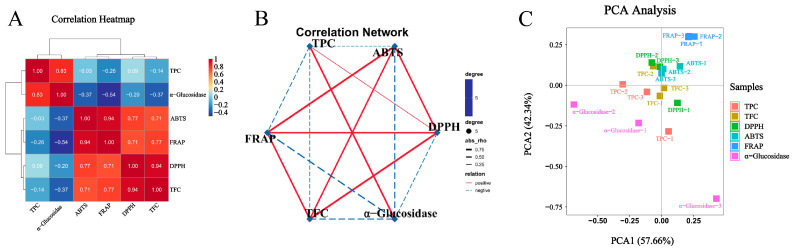
Associations among different indicators. (**A**) Correlation heatmap. (**B**) Correlation network. (**C**) Principal component analysis, the numbers 1, 2, and 3 represent the three replicates of the sample. TPC = total phenolic content; TFC = total flavonoid content; DPPH = DPPH·scavenging capacity; ABTS = ABTS·^+^ scavenging capacity; FRAP = ferric reducing antioxidant power; α-Glucosidase = α-glucosidase inhibition capacity.

**Figure 3 antioxidants-12-01505-f003:**
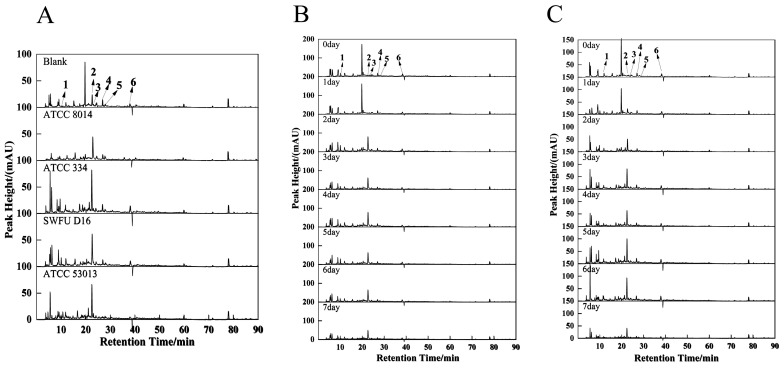
HPLC chromatogram of the FGPL. 1—gallic acid, 2—catechin, 3—chlorogenic acid, 4—epicatechin, 5—dihydromyricetin and 6—epicatechin gallate. (**A**) Liquid chromatogram of FGPL fermentation broth of different strains at 280 nm. (**B**) Liquid chromatography of FGPL SWFU D16 fermentation broth at 280 nm at different fermentation stages. (**C**) Liquid chromatogram of FGPL ATCC 334 fermentation broth at 280 nm at different fermentation stages.

**Figure 4 antioxidants-12-01505-f004:**
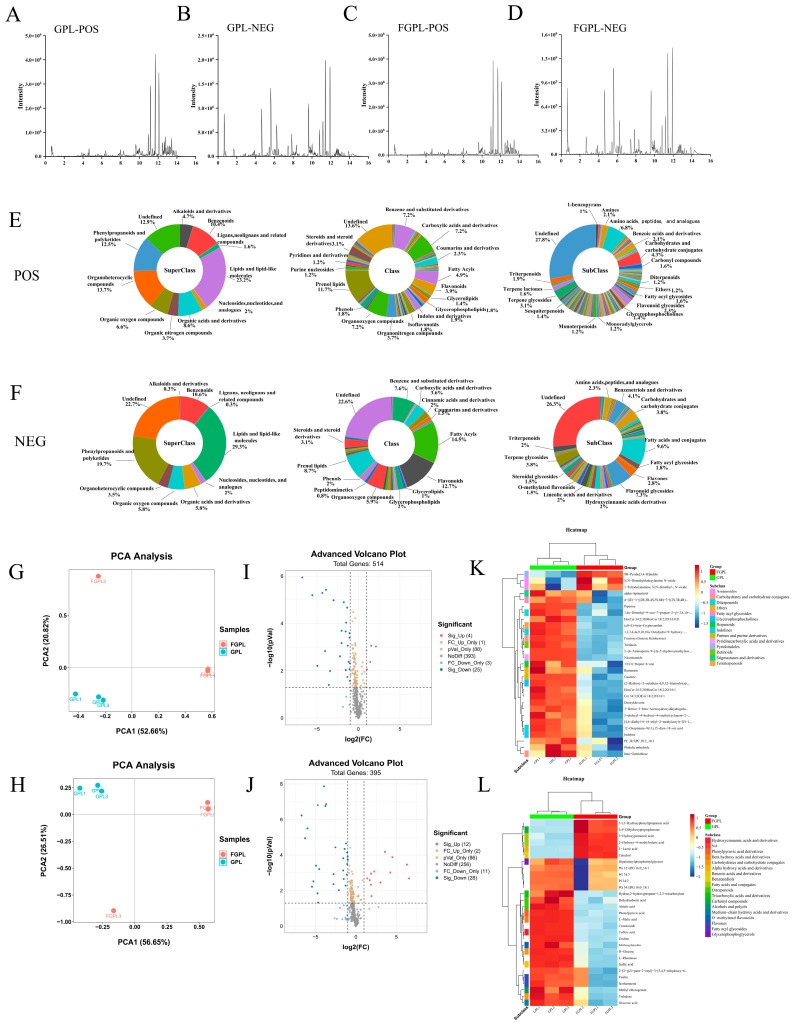
BPC of GPL and FGPL in the positive (**A**,**C**) and negative (**B**,**D**) ion modes. Mean ± standard deviation (n = 3). The schematic diagram of the different classifications of the metabolites of FGPL (**E**,**F**). PCA, volcano plots, and a heat map showed the metabolite changes in GPL and FGPL. (**G**) Score plot in positive ion mode. (**H**) Score plot in negative ion mode. (**I**) Volcano plot in positive ion mode. (**J**) Volcano plot in negative ion mode. The blue dots represent significantly downregulated and differentially expressed metabolites. The red dots represent significantly upregulated and differentially expressed metabolites. Significant metabolite differences between groups were determined by *p <* 0.05 and an absolute *fold change ≥* 1. (**K**) Heat map in positive ion mode. (**L**) Heat map in negative ion mode. Each sample is represented by one column, and each metabolite is visualized in one row. Red indicates high abundance; blue indicates relatively low metabolite abundance.

**Figure 5 antioxidants-12-01505-f005:**
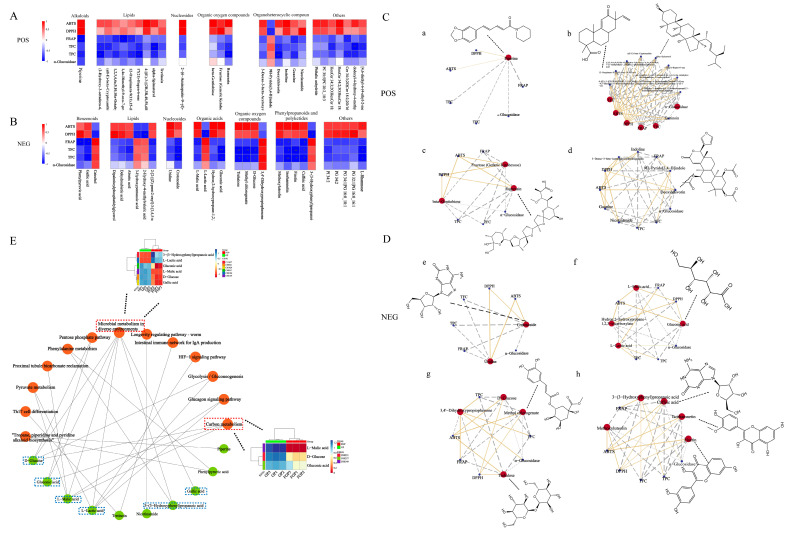
Spearman’s analysis and associated network diagram show the correlation between metabolites and five functional activities. (**A**,**B**) Spearman’s analysis of FGPL in the positive and negative ion modes. (**C**,**D**) Associated networks of FGPL in positive and negative ion modes (a–h). Asterisks represent *p <* 0.05 *, *p ≤* 0.01 **, respectively. The gray line represents a negative correlation, and the yellow line represents a positive correlation. (**E**) KEGG pathway network diagram. The orange elliptical nodes represent pathways, and the green elliptical nodes represent metabolites.

**Table 1 antioxidants-12-01505-t001:** Change in the six compounds during different fermentation stages for different probiotics by HPLC.

		Fermentation Time (Days)							
Compounds	Probiotics	0	1	2	3	4	5	6	7
gallic acid	Blank	ND	ND	ND	ND	ND	ND	ND	ND
	ATCC 8014	ND	ND	0.46 ± 0.06 ^a^	0.21 ± 0.0374 ^b^	0.09 ± 0.0395 ^c^	ND	ND	ND
	ATCC 334	0.27 ± 0.0302 ^abc^	0.16 ± 0.0489 ^bc^	0.17 ± 0.0509 ^bc^	0.2 ± 0.0369 ^ab^	0.18 ± 0.0332 ^abc^	0.24 ± 0.0443 ^a^	0.17 ± 0.019 ^c^	ND
	SWFU D16	0.65 ± 0.0575 ^b^	0.14 ± 0.0124 ^f^	0.78 ± 0.069 ^a^	0.54 ± 0.0478 ^b^	0.41 ± 0.0363 ^c^	0.36 ± 0.0319 ^cd^	0.29 ± 0.0257 ^de^	0.24 ± 0.0212 ^ef^
	ATCC 53013	ND	ND	ND	ND	ND	ND	ND	ND
catechin	Blank	0.147 ± 0.026 ^ab^	0.149 ± 0.0087 ^ab^	0.1671 ± 0.037 ^ab^	0.1675 ± 0.042 ^ab^	0.1711 ± 0.03 ^ab^	0.1691 ± 0.01 ^ab^	0.1699 ± 0.038 ^a^	0.146 ± 0.0082 ^b^
	ATCC 8014	0.98 ± 0.397 ^b^	0.21 ± 0.009 ^b^	0.23 ± 0.0327 ^b^	3.82 ± 0.278 ^a^	0.37 ± 0.081 ^b^	3.63 ± 0.296 ^a^	0.18 ± 0.036 ^b^	0.21 ± 0.058 ^b^
	ATCC 334	ND	0.09 ± 0.0165 ^c^	0.39 ± 0.0716 ^b^	0.65 ± 0.119 ^a^	0.41 ± 0.0753 ^b^	0.59 ± 0.108 ^ab^	0.48 ± 0.0881 ^ab^	0.05 ± 0.009 ^c^
	SWFU D16	0.32 ± 0.345 ^c^	0.32 ± 0.0285 ^c^	6.69 ± 0.437 ^a^	5.25 ± 0.343 ^b^	6.09 ± 0.397 ^a^	5.57 ± 0.363 ^ab^	5.53 ± 0.361 ^ab^	4.29 ± 0.28 ^ab^
	ATCC 53013	0.53 ± 0.047 ^b^	0.37 ± 0.033 ^c^	0.51 ± 0.045 ^b^	0.66 ± 0.058 ^a^	0.33 ± 0.0229 ^c^	0.42 ± 0.037 ^bc^	0.33 ± 0.029 ^c^	0.31 ± 0.027 ^c^
chlorogenic acid	Blank	0.0057 ± 0.0026 ^b^	0.0061 ± 0.001 ^b^	0.0199 ± 0.0042 ^a^	0.0196 ± 0.0064 ^a^	0.0211 ± 0.0054 ^a^	0.0217 ± 0.0023 ^a^	0.0185 ± 0.0047 ^a^	0.0039 ± 0.0007 ^b^
	ATCC 8014	0.59 ± 0.0884 ^a^	0.041 ± 0.00614 ^b^	0.08 ± 0.012 ^b^	0.04 ± 0.006 ^b^	0.05 ± 0.0075 ^b^	0.08 ± 0.012 ^b^	0.11 ± 0.0165 ^b^	0.09 ± 0.0135 ^b^
	ATCC 334	0.1 ± 0.0148 ^b^	0.09 ± 0.0133 ^b^	0.09 ± 0.004 ^b^	0.11 ± 0.0163 ^b^	0.08 ± 0.0119 ^b^	0.16 ± 0.0237 ^a^	0.18 ± 0.0267 ^a^	0.03 ± 0.0046 ^c^
	SWFU D16	0.11 ± 0.0259 ^ab^	0.11 ± 0.0244 ^ab^	0.16 ± 0.0378 ^a^	0.08 ± 0.0189 ^b^	0.12 ± 0.0284 ^ab^	0.1 ± 0.0236 ^b^	0.09 ± 0.0213 ^b^	0.076 ± 0.018 ^b^
	ATCC 53013	0.11 ± 0.0072 ^a^	0.093 ± 0.0061 ^bc^	0.103 ± 0.0067 ^ab^	0.084 ± 0.0055 ^c^	0.056 ± 0.0037 ^d^	0.09 ± 0.0059 ^bc^	0.059 ± 0.39 ^d^	0.064 ± 0.0042 ^d^
epicatechin	Blank	0.0023 ± 0.00054 ^b^	0.0068 ± 0.0016 ^b^	0.102 ± 0.024 ^a^	0.0089 ± 0.0021 ^b^	0.0094 ± 0.0022 ^b^	ND	ND	ND
	ATCC 8014	0.08 ± 0.0391 ^b^	0.04 ± 0.339 ^b^	0.22 ± 0.0105 ^b^	0.57 ± 0.0753 ^a^	0.03 ± 0.0138 ^b^	ND	0.008 ± 0.0731 ^b^	0.004 ± 0.005 ^b^
	ATCC 334	0.78 ± 0.123 ^a^	0.73 ± 0.115 ^ab^	0.58 ± 0.0911 ^b^	0.066 ± 0.0104 ^c^	0.03 ± 0.0047 ^c^	0.06 ± 0.0094 ^c^	0.05 ± 0.078 ^c^	ND
	SWFU D16	0.91 ± 0.0727 ^ab^	0.94 ± 0.0751 ^a^	0.84 ± 0.0671 ^ab^	0.51 ± 0.0407 ^cd^	0.78 ± 0.0623 ^b^	0.77 ± 0.0615 ^b^	0.59 ± 0.0471 ^c^	0.39 ± 0.0311 ^d^
	ATCC 53013	0.025 ± 0.0059 ^c^	0.033 ± 0.0078 ^c^	0.03 ± 0.0071 ^c^	0.022 ± 0.0052 ^c^	0.078 ± 0.018 ^c^	0.003 ± 0.0007 ^c^	0.28 ± 0.066 ^b^	0.56 ± 0.132 ^a^
dihydromy- ricetin	Blank	ND	0.0088 ± 0.0007 ^b^	0.0075 ± 0.0006 ^b^	0.0064 ± 0.00051 ^b^	0.143 ± 0.0114 ^a^	0.0086 ± 0.0007 ^b^	ND	ND
	ATCC 8014	0.07 ± 0.0099 ^bc^	0.51 ± 0.114 ^a^	0.13 ± 0.051 ^b^	0.05 ± 0.021 ^c^	ND	0.007 ± 0.0041 ^c^	0.07 ± 0.0217 ^bc^	0.06 ± 0.0103 ^bc^
	ATCC 334	0.02 ± 0.00481 ^bc^	0.008 ± 0.00192 ^c^	0.17 ± 0.0409 ^a^	0.04 ± 0.00626 ^bc^	ND	0.04 ± 0.00498 ^bc^	0.06 ± 0.0693 ^b^	ND
	SWFU D16	0.05 ± 0.0076 ^a^	0.034 ± 0.0057 ^bc^	0.029 ± 0.0088 ^bc^	0.026 ± 0.0054 ^c^	0.046 ± 0.0023 ^ab^	ND	0.008 ± 0.0004 ^d^	ND
	ATCC 53013	0.027 ± 0.0022 ^b^	0.032 ± 0.0026 ^ab^	0.033 ± 0.0026 ^ab^	0.034 ± 0.0027 ^a^	0.019 ± 0.0015 ^c^	0.031 ± 0.0025 ^ab^	ND	ND
epicatechin gallate	Blank	ND	0.146 ± 0.0074 ^a^	0.156 ± 0.0079 ^a^	0.102 ± 0.0051 ^b^	0.0055 ± 0.0003 ^c^	ND	ND	ND
	ATCC 8014	ND	0.88 ± 0.246 ^a^	ND	ND	ND	0.76 ± 0.107 ^a^	0.07 ± 0.018 ^b^	0.07 ± 0.016 ^b^
	ATCC 334	0.14 ± 0.0427 ^cd^	0.12 ± 0.0366 ^cd^	0.08 ± 0.0244 ^d^	0.33 ± 0.101 ^ab^	0.22 ± 0.067 ^bc^	0.36 ± 0.109 ^a^	0.12 ± 0.0366 ^cd^	0.05 ± 0.0153 ^d^
	SWFU D16	0.15 ± 0.0128 ^bc^	0.17 ± 0.0291 ^ab^	0.28 ± 0.0482 ^a^	0.17 ± 0.0359 ^a^	0.12 ± 0.0173 ^c^	0.19 ± 0.0246 ^ab^	0.14 ± 0.01 ^bc^	0.08 ± 0.0247 ^c^
	ATCC 53013	ND	0.26 ± 0.013 ^a^	0.18 ± 0.0091 ^b^	ND	ND	0.049 ± 0.0025 ^c^	ND	ND

ND = not detected. Different lower-case letters in the same row indicate significant differences at *p <* 0.05.

## Data Availability

The data are contained within the article or in the Appendix A.

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
