# Peer review of "Chemical Constituents, Antioxidant, and α-Glucosidase Inhibitory Activities of Different Fermented Gynostemma Pentaphyllum Leaves and Untargeted Metabolomic Measurement of the Metabolite Variation"

_antioxidants, 2023, doi:10.3390/antiox12081505_

Round 1

Reviewer 1 Report

This manuscript reports on the study of the changes in chemical composition, antioxidant and α-glucosidase inhibitory activities of the leaves of the Chinese edible and medicinal plant Gynostemma pentaphyllum in the process of fermentation with four strains of probiotic bacteria. The methods applied are up-to date and appropriate, the conclusions are supported by the results.

There are however several points which should be addressed by the Authors, as follows:

1.       To subsection 2.2., description of fermentation and sample preparation should be added to the title.

2.       Subsection 2.10. Information about the standards is lacking.

3.       Subsection 2.11.: Information about the standards is lacking.

4.       Line 251, the reference is not in the format of the Journal.

5.       Line 278, Fig.2C: What is the meaning of the numbers (1,2,3) on the PCA diagram?

6.       Subsection 3.6., the first sentence of paragraph 2: How were these 909 metabolites identified? It is necessary do describe this in Experimental.

7.       Some explanation of KEGG pathway analysis is necessary

8.       Lines 426 – 427: carbon metabolism can be microbial, too. Please, explain.

This manuscript reports on the study of the changes in chemical composition, antioxidant and α-glucosidase inhibitory activities of the leaves of the Chinese edible and medicinal plant Gynostemma pentaphyllum in the process of fermentation with four strains of probiotic bacteria. The methods applied are up-to date and appropriate, the conclusions are supported by the results.

There are however several points which should be addressed by the Authors, as described in the Comments to Authors.

Author Response

Dear Reviewer:

We are grateful to have been given the opportunity to revise the manuscript entitled "Chemical constituents, antioxidant and α-glucosidase inhibitory activities of different fermented Gynostemma Pentaphyllum leaves and untargeted metabolomic measurement of the metabolite variation" (Manuscript ID: 2502079). We thank your affirmative evaluation and constructive comments on our manuscript, and those comments are all valuable and helpful for improving our paper. The manuscript has now been revised in line with all of the review comments, and our native-speaker colleague also carefully edited the manuscript. The revised portions are modified and marked in red in the paper. The main corrections in the paper and the responses to the reviewer’s comments are as follows:

  1. To subsection 2.2., description of fermentation and sample preparation should be added to the title.

Response: Thank you for your advice. We have modified the title of section 2.2 to "Fermentation and Sample Preparation.". Please see line 114.

  1. Subsection 2.10. Information about the standards is lacking.

Response: Thank you for pointing it out. We have added information about standards to the manuscript. Please see line 188-190, 198-199.

  1. Subsection 2.11.: Information about the standards is lacking.

Response: We admire the reviewer’s rich experience in mass spectrometry analysis and identification, and are grateful for your thorough and insightful review. In this study, the untargeted metabolomics analysis was performed by APPLIED PROTEIN technology Co., Ltd (www.aptbiotech.com). Based on a large number of pure standards and commercial databases, this company has developed a high-quality in-house database, including more than 30, 000 substances. By matching with the molecular mass (molecular mass error within <10 ppm), secondary fragmentation spectrum, retention time and other information of metabolites in the database, the structure of metabolites in biological samples was identified, and the results were checked and confirmed. For the metabolites not matched to standard substances, they were matched to public databases or manually identified based on their mass spectral cleavage rules.

According to the reviewer´s suggestion, we add a description of how to identify metabolites in the manuscript "3.6 Metabolomic Analysis". Please see line 393-396. We hope these modifications meet your requests and thanks again to the reviewers.

  1. Line 251, the reference is not in the format of the Journal.

Response: Thank you for the reminder. We have revised it in the manuscript, as well as to another reference that did not meet the requirements of the journal citation format. Please see line 264 and 269.

  1. Line 278, Fig.2C: What is the meaning of the numbers (1, 2, 3) on the PCA diagram?

Response: Thanks. The numbers 1, 2, and 3 in the PCA diagram represent the three replicates of the sample. We added descriptions of numbers 1, 2, 3 in the PCA diagram title. Please see line 331-332, Fig.2C.

  1. Subsection 3.6., the first sentence of paragraph 2: How were these 909 metabolites identified? It is necessary do describe this in Experimental.

Response: Thank you for your advice. Through the third recommendation above, we have described in detail how to identify metabolites in the manuscript.

In addition to comparing with reference standards and commercial databases, we primarily analyze based on the mass spectral cleavage rules of the substances. For example, the secondary fragmentation of flavonoids has strong regularity. It is well known that the fragments of flavonoid glycosides are usually neutral loss glycosides, and the flavonol core will appear [M-1]- and [M-2]-ion peaks under negative ions mode. In addition, C-glycosides (e.g isosaponarin) have many frequent fragments such as M-30, M-60, M-90, M-120, etc. Specifically, the compounds with kaempferol as the mother nucleus have a characteristic fragment (287.0550) in positive ion mode, while in negative ion mode, the characteristic fragments are 285.0404 and 284.0326. If the mother nucleus of the compound is isorhamnetin, the characteristic fragments are 317.0655 (positive ion mode), 315.0510 and 314.0432 (negative ion mode). For compounds with the quercetin-based mother nucleus, there are 303.0499 (positive ion mode), 301.0353 and 300.0421 (negative ion mode). While 275.0914 and 273.0768 are, respectively, the characteristic fragments of compounds with phloretin-based mother nucleus in positive and negative ion mode. Additionally, the same method was also used for the identification of other metabolites. We hope these modifications meet your requests and thanks again to the reviewers.

  1. Some explanation of KEGG pathway analysis is necessary

Response: Thank you for pointing it out. We have added descriptions of microbial metabolism in diverse environments and carbon metabolism to the manuscript. Please see line 448-456.

  1. Lines 426 – 427: carbon metabolism can be microbial, too. Please, explain.

Response: Thank you for pointing it out and we sincerely admire the level of reviewers. The carbon metabolism pathway is a crucial aspect of microbial metabolism in probiotic fermentation. In probiotic fermentation, microorganisms metabolize sugars and carbohydrates through the carbon metabolism pathway to produce energy and generate metabolites [1]. Meanwhile, as the most important basic metabolism in plants, carbon metabolism can provide essential energy for plant life activities, The carbon metabolic pathway of plants regulates plant growth, development and adaptation to various environments through photosynthesis, glycolysis pathway, TCA cycle and other metabolic pathways [2]. We have added a description of carbon metabolism in microorganisms to the manuscript. We hope these modifications meet your requests and thanks again to the reviewers. Please see line 453-456.

Reference

[1] Schada von Borzyskowski L, Bernhardsgrütter I, Erb T J. Biochemical unity revisited: microbial central carbon metabolism holds new discoveries, multi-tasking pathways, and redundancies with a reason[J]. Biological Chemistry, 2020, 401(12): 1429-1441. https://doi.org/10.1515/hsz-2020-0214.

[2] Yu J, Li R, Fan N, et al. Metabolic pathways involved in carbon dioxide enhanced heat tolerance in bermudagrass[J]. Frontiers in plant science, 2017, 8: 1506. https://doi.org/10.3389/fpls.2017.01506.

Thank you and best regards.

Sincerely yours,

Prof. Dr. Xuechun Zhang

Southwest Forestry University, Kunming, 650224, China

E-mail address: [email protected]

Reviewer 2 Report

This manuscript described chemical constituents, antioxidant and α-glucosidase inhibitory activities of different fermented Gynostemma pentaphyllum leaves and untargeted metabolomic measurement of the metabolite variation. The manuscript was well written and very interesting and the methods for experiment are classic and logical. However, there are some critical issues to be revised before the publication.

1) Gynostemma pentaphyllum is scientific name of plant, and it should be italic in the whole manuscript.

2) In abstract, the conclusion is very ambiguous. You need to be specific about your results and what you want to say as a result.

3) Overall, it would be nice if all the figures could be modified to make them easier to read, such as increasing the resolution and repositioning the text so that it doesn't overlap on the graph.

4) On line 183, we need to change 3-18% to 13-18%

5) In the 2.10 section, you need more information about the columns to know which ones you used.

6) In the 2.11 section, you need more information about the columns to know which C-18 columns you used.

7) On line 220, PH -> pH

8) Within Figure 1, it should be mentioned what the letters at the top of each bar graph mean.

9) For the Figure 1 graphs, it would be nice to have a unified y-axis range for b1 through b4 and c1 through c4.

10) In Figure 3, Method is 90 min, but the HPLC chromatogram in Figure 3 is only up to 50 min. I would like to have the HPLC chromatogram raw data up to 90 min.

11) It would be nice to have HPLC chromatograms of the flavonoids and phenolics standards Gallic acid, Catechins, Chlorogenic acid, Epicatechin, Dihydromyricetin, Epicatechin gallate mentioned in Supply Table 1 at RT under the HPLC conditions listed in the Method.

12) Within Table 1, I think we need to uniformly modify the alphabetization so that the numbers are superscripted in batches.

13) The text in Figure 4 is too small to read – It would be better to increase the resolution of the image or arrange the image differently to make the data more visible.

14) Figure 5 would be also better to increase the resolution of the image to make the data more visible.

15) It would be better to add similar previous experiment using other sources for comparing the results in the study in the discussion section.

Extensive editing of English language required

Author Response

Dear Reviewer:

We are grateful to have been given the opportunity to revise the manuscript entitled "Chemical constituents, antioxidant and α-glucosidase inhibitory activities of different fermented Gynostemma Pentaphyllum leaves and untargeted metabolomic measurement of the metabolite variation" (Manuscript ID: 2502079). We thank your affirmative evaluation and constructive comments on our manuscript, and those comments are all valuable and helpful for improving our paper. The manuscript has now been revised in line with all of the review comments, and our native-speaker colleague also carefully edited the manuscript. The revised portions are modified and marked in red in the paper. The main corrections in the paper and the responses to the reviewer’s comments are as follows:

  1. Gynostemma pentaphyllum is scientific name of plant, and it should be italic in the whole manuscript.

Response: Thank you for the reminder. We have revised the whole manuscript.

  1. In abstract, the conclusion is very ambiguous. You need to be specific about your results and what you want to say as a result.

Response: Thank you for your advice. We have supplemented the description of the results in the abstract. We hope these modifications meet your requests and thanks again to the reviewers.

  1. Overall, it would be nice if all the figures could be modified to make them easier to read, such as increasing the resolution and repositioning the text so that it doesn't overlap on the graph.

Response: Thank you for your advice. Our original Word version was clear, but it may have become less clear in the PDF format due to Word conversion. We have modified the original images and will upload the high-resolution PDF manuscript in the subsequent revision. We hope these modifications meet your requests and thanks again to the reviewers.

  1. On line 183, we need to change 3-18% to 13-18%.

Response: Thank you for pointing it out. We have revised it in the manuscript. Please see line 192.

  1. In the 2.10 section, you need more information about the columns to know which ones you used.

Response: Thank you for your advice. We have added more detailed information in the manuscript to describe the experimental conditions. Please see line 188-190, 198-199.

  1. In the 2.11 section, you need more information about the columns to know which C-18 columns you used.

Response: Thank you for your advice. We have added more detailed information in the manuscript to describe the experimental conditions. Please see line 206.

  1. On line 220, PH -> pH

Response: Thank you for pointing it out. We have corrected it in the manuscript. Please see line 233.

  1. Within Figure 1, it should be mentioned what the letters at the top of each bar graph mean.

Response: Thank you for pointing it out. We have added this information in the title to Figure 1. Please see line 323-328.

  1. For the Figure 1 graphs, it would be nice to have a unified y-axis range for b1 through b4 and c1 through c4.

Response: Thank you for your advice. We have modified all the y-axes in Figure 1.

  1. In Figure 3, Method is 90 min, but the HPLC chromatogram in Figure 3 is only up to 50 min. I would like to have the HPLC chromatogram raw data up to 90 min.

Response: Thank you very much for your advice. We have modified the HPLC and extended it to 90 min. Please see Figure 3, We hope these modifications meet your requests and thanks again to the reviewers.

  1. It would be nice to have HPLC chromatograms of the flavonoids and phenolics standards Gallic acid, Catechins, Chlorogenic acid, Epicatechin, Dihydromyricetin, Epicatechin gallate mentioned in Supply Table 1 at RT under the HPLC conditions listed in the Method.

Response: We admire the reviewer’s rich experience in HPLC, and are grateful for your thorough and insightful review. In fact, we conducted individual HPLC analysis on 17 different standard substances in the preliminary stage. By comparing parameters such as retention time and peak area of each standard substance, we ultimately identified the standard substances suitable for Gynostemma pentaphyllum, including Gallic acid, Catechins, Chlorogenic acid, Epicatechin, Dihydromyricetin, and Epicatechin gallate. However, due to space constraints, we are unable to present the individual chromatograms of these substances in the manuscript. In this response, we are providing the retention times and standard curve equations for these 17 substances for the reviewer's reference, hoping to satisfy the reviewer. Thanks again to the reviewers.

The standard curve of flavonoids and phenolics

Number

Wavelength/nm

Compounds

Retention time/min

standard curve

1

280

Gallic acid

11.507

y = 662.72x + 73.856 (R² = 0.9995)

2

280

Catechins

22.861

y = 185.86x - 26.949 (R² = 0.9991)

3

280

Epicatechin

27.1776

y = 213.05x + 16.625 (R² = 0.9983)

4

280

Dihydromyricetin

28.7668

y = 446.13x + 23.478 (R² = 0.9994)

5

280

Epicatechin gallate

37.3718

y = 413.08x + 89.615 (R² = 0.9954)

6

310

chlorogenic acid

23.0426

y = 336.05x - 1.1845 (R² = 0.9998)

7

310

Rosmarinic acid

46.1882

y = 274.64x - 27.312 (R² = 0.9993)

8

310

Baicalin

49.8192

y = 324.12x - 1.6326 (R² = 0.9974)

9

310

Hesperidin

60.9912

y = 247.1x + 15.53 (R² = 0.9992)

10

340

Rutin

36.8364

y = 97.66x + 0.2391 (R² = 0.9985)

11

340

Ferulic acid

36.933

y = 484.56x + 1.9186 (R² = 0.9997)

12

340

Resveratrol

49.6478

y = 289.2x - 5.31 (R² = 0.9998)

13

340

Apigenin

59.8022

y = 472.77x - 0.899 (R² = 0.9993)

14

340

Baicalein

61.7676

y = 239.15x - 31.549 (R² = 0.9985)

15

360

Myricetin

46.1408

y = 226.05x - 7.4156 (R² = 0.9965)

16

360

Luteolin

54.95

y = 30.244x - 36.311 (R² = 0.9985)

17

360

Quercetin

55.3762

y = 264.13x + 8.0285 (R² = 0.9995)

  1. Within Table 1, I think we need to uniformly modify the alphabetization so that the numbers are superscripted in batches.

Response: Thank you for pointing it out. We have corrected it in the manuscript. Please see Table 1.

  1. The text in Figure 4 is too small to read – It would be better to increase the resolution of the image or arrange the image differently to make the data more visible.

Response: Thank you for your advice. We have now revised the font size and re-uploaded a high-resolution PDF version, hoping to satisfy the reviewer. Thanks again to the reviewers.

  1. Figure 5 would be also better to increase the resolution of the image to make the data more visible.

Response: Thanks. We have uploaded the modified original images in the revised manuscript. Thanks again to the reviewers.

  1. It would be better to add similar previous experiment using other sources for comparing the results in the study in the discussion section.

Response: Thank you for your advice. We have added literature in the discussion section that presents similar results. We hope these modifications meet your requests and thanks again to the reviewers.

References

[17] Ru, Y.R.; Wang, Z.X.; Li, Y.J.; Kan, H.; Kong, K.W.; Zhang, X.C. The influence of probiotic fermentation on the active compounds and bioactivities of walnut flowers. J Food Biochem 2022, 46, e13887, doi:10.1111/jfbc.13887.

[25] Wang, Y.-R.; Xing, S.-F.; Lin, M.; Gu, Y.-L.; Piao, X.-L. Determination of flavonoids from Gynostemma pentaphyllum using ultra-performance liquid chromatography with triple quadrupole tandem mass spectrometry and an evaluation of their antioxidant activity in vitro. J Liq Chromatogr R T 2018, 41, 437-444, doi:10.1080/10826076.2018.1448281.

Thank you and best regards.

Sincerely yours,

Prof. Dr. Xuechun Zhang

Southwest Forestry University, Kunming, 650224, China

E-mail address: [email protected]

Reviewer 3 Report

Criticism of the Content:

The abstract of the manuscript summarizes the methods used during the project and the results. The introductory part also carries the necessary information for the introduction of the tests and sufficiently helps the understanding of the project.

My comments refer to the Methods and Results and Discussion sections:

-        2.3. Fermentation pH: it was determined every 24 hours? It was performed after the centrifugation process? Just on Figure 1 a1-a4 we can check it! Please complete the method.

-        What do they mean in the first diagram (Figure 1), e.g. the mark a1-a4, the numbers represent the individual fermenting strains? If so, which one?

-        In the same figure (Figure 1), what are the meanings of a-b-c-d and the statistical symbols ab, bc, etc.?

-        The resolution of many figures is not suitable because the text is not visible on them, and the placement of the text is not suitable because it is covered. In addition, the figures are extremely crowded, e.g. Figures 2, 4, 5.

-        Figure 4 is unfortunately unintelligible for this reason, but the legends on Figure 5 are also not appropriate in some places (A,B,E).

Formal criticism:

There are a lot of typing and formatting errors: Line 89, 95-96, 114, 171, 281, 299, 282, 423.

Author Response

Dear Reviewer:

We are grateful to have been given the opportunity to revise the manuscript entitled "Chemical constituents, antioxidant and α-glucosidase inhibitory activities of different fermented Gynostemma Pentaphyllum leaves and untargeted metabolomic measurement of the metabolite variation" (Manuscript ID: 2502079). We thank your affirmative evaluation and constructive comments on our manuscript, and those comments are all valuable and helpful for improving our paper. The manuscript has now been revised in line with all of the review comments, and our native-speaker colleague also carefully edited the manuscript. The revised portions are modified and marked in red in the paper. The main corrections in the paper and the responses to the reviewer’s comments are as follows:

2.3. Fermentation pH: it was determined every 24 hours? It was performed after the

centrifugation process? Just on Figure 1 a1-a4 we can check it! Please complete the method.

Response: Thank you for your advice. The fermentation broth is collected once every 24 hours and then centrifuged at 5000 × g for 10 min at 25°C. Finally, the supernatant is collected for analysis. We have added a detailed description of the determination of pH in the "2.3 Fermentation pH" section of the manuscript. Please see line 129-131.

- What do they mean in the first diagram (Figure 1), e.g. the mark a1-a4, the numbers represent the individual fermenting strains? If so, which one?

Response: Thank you for pointing it out. The labels a1-g4 represent the identification numbers of each indicator chart, and the following numbers 1, 2, 3, 4 correspond to the following strains: 1) ATCC 8014 (L. plantarum), 2) ATCC 334 (L. casei), 3) SWFU D16 (L. plantarum), and 4) ATCC 53013 (L. rhamnosus). We have added a detailed description in the Figure 1 title. Please see line 323-326, and thanks again to the reviewers.

- In the same figure (Figure 1), what are the meanings of a-b-c-d and the statistical symbols ab, bc, etc.?

Response: Thank you for pointing it out. The symbols a-b-c-d and ab, bc, etc. in Figure 1 represent statistical significance analysis. We have added a detailed description of them in the title of Figure 1. Please see line 326-328. Thanks again to the reviewers.

- The resolution of many figures is not suitable because the text is not visible on them, and the placement of the text is not suitable because it is covered. In addition, the figures are extremely crowded, e.g. Figures 2, 4, 5.

Response: Sincere thanks to the reviewers. Our original word version was clear, but it may have become less clear in the PDF format due to Word conversion. We have now revised the font size and re-uploaded a high-resolution PDF version, hoping to satisfy the reviewer. Thanks again to the reviewers.

- Figure 4 is unfortunately unintelligible for this reason, but the legends on Figure 5 are also not appropriate in some places (A, B, E)

Response: Thank the reviewers for their patient advice. We have uploaded the modified original images in the revised manuscript. Thanks again to the reviewers.

We have tried our best to improve the manuscript, and these changes will not influence the content and framework of the paper.

We appreciate for Editors/Reviewers’ warm work earnestly and hope that the correction will meet with approval.

Once again, thanks very much for your comments and suggestions.

Thank you and best regards.

Sincerely yours,

Prof. Dr. Xuechun Zhang

Southwest Forestry University, Kunming, 650224, China

E-mail address: [email protected]

Round 2

Reviewer 1 Report

The Authors answered most questions satisfactorily. There are only a couple of points needing their attention:

1.       Line 224: Databases used for compound identification should be mentioned here

2.       Line 393: You should add a text to Supplementary material, and refer here to it. This text has to include point 3 of your answered.

Acceptable

Author Response

Dear Reviewers:

The manuscript has now been revised in line with all of the review comments, and our native-speaker colleague also carefully edited the manuscript. The revised portions are modified and marked in red in the paper. The main corrections in the paper and the responses to the reviewer’s comments are as follows:

  1. Line 224: Databases used for compound identification should be mentioned here.

Response: Thank the reviewers for their careful guidance. We have added information about the database to the manuscript. Please see line 226-228.

  1. Line 393: You should add a text to Supplementary material, and refer here to it. This text has to include point 3 of your answered.

Response: Thank you for the reminder. Due to the limitations of the Excel spreadsheet, we have provided a separate Word document as a supplementary document for metabolite identification. We hope this will satisfy the reviewers and would like to express our gratitude once again.

Sincerely yours,

Prof. Dr. Xuechun Zhang

Southwest Forestry University, Kunming, 650224, China

E-mail address: [email protected]

Reviewer 2 Report

All the comments were well-addressed. 

Minor editing of English language required

Author Response

Dear Reviewers:

The manuscript has now been revised in line with all of the review comments, and our native-speaker colleague also carefully edited the manuscript. The revised portions are modified and marked in blue in the paper.

1 Minor editing of English language required.

Response: We have thoroughly edited the manuscript, correcting the grammar errors that were present. We appreciate the careful guidance of the reviewers.

Sincerely yours,

Prof. Dr. Xuechun Zhang

Southwest Forestry University, Kunming, 650224, China

E-mail address: [email protected]